# Factors Influencing Consumers’ Purchase Intention on Cold Chain Aquatic Products under COVID-19: An Investigation in China

**DOI:** 10.3390/ijerph19084903

**Published:** 2022-04-18

**Authors:** Xin Shen, Xun Cao, Sonia Sadeghian Esfahani, Tayyaba Saleem

**Affiliations:** 1College of Economics and Management, Shanghai Ocean University, Shanghai 201306, China; xshen@shou.edu.cn (X.S.); tayyabasaleem31@yahoo.com (T.S.); 2College of Science and Engineering, University of Tasmania, Hobart, TAS 7001, Australia; sonia.sadeghianesfahani@utas.edu.au

**Keywords:** cold chain aquatic products, health concern, the theory of planned behavior, structural equation model

## Abstract

Following the detection of COVID-19 in cold chain aquatic products (CCAP) at Xinfadi Produce Wholesale Market in Beijing, China, in June 2020, novel coronavirus positive tests of CCAP have been reported in such markets of Dalian, Xi’an, Qingdao, Taiyuan, and other places, which has aroused the concern of Chinese consumers. The CCAP outbreak puts tremendous pressure on public health management and threatens customer well-being. This article uses the theoretical model of planned behavior (TPB) to investigate Chinese consumers’ purchasing intentions of CCAP under this circumstance. A total of 783 questionnaires were administered in China with empirical analysis through a structural equation model. The results show that attitudes (ATT) towards the safety of CCAP and subjective norms (SN) have significant positive effects on customers’ purchasing behavior intention (BI); the emotional response to the health concern (EM) of CCAP has a significant positive impact on SN, ATT, and BI; and BI of CCAP is significantly affected by individual characteristics. The BI of CCAP for those married consumers living in cities and towns with a low monthly consumption frequency is more likely to be affected by the new coronavirus epidemic events. This paper is one of the first studies that contribute to the literature by exploring the influencing factors of the consumption behavior intention over the COVID-19 pandemic in China from a public health perspective. The findings provide significant implications for cold chain operators, market managers, and policymakers to develop guidelines and offer a framework to initiate and support the produce market and boost consumer health confidence in CCAP at the practitioner level.

## 1. Introduction

China is a top aquaculture producer and a significant consumer of aquatic products globally. The number of marine products consumed per capita in China rose from 14.4 kg in 1993 to 41 kg in 2018 [1]. China is predicted to account for 38% of global fish consumption by 2030 [2]. However, since 2020, COVID-19 has swept the globe and incidents related to aquatic product epidemics have occurred frequently; therefore, consumers have expressed concern about whether consuming aquatic products will affect their health. On 11 June 2020, an epidemic broke out at the Beijing Xinfadi Agricultural Products Wholesale Market. The researchers revealed that the source of the virus in this outbreak was most likely imported salmon from a high-incidence area outside China and suggested that cold chain transport could be a novel coronavirus transmission route [3]. Although the official media has stated that salmon does not carry COVID-19, consumers still believe that buying and eating salmon can affect their health. Subsequent events have heightened those concerns. On 22 July 2020, Dalian City successively found confirmed cases of novel coronavirus pneumonia and most patients had a history of activities at Dalian Kaiyang Seafood Co., Ltd. in Liaoning, China, or a history of contact with relevant personnel. In addition, on 13 August 2020, some frozen vannamei shrimp packaging samples at Xi’an Fangxin Seafood Market showed positive results of the new coronavirus nucleic acid test. On 24 September 2020, two workers of Qingdao Dagang Company who were loading and unloading frozen seafood were infected with the new coronavirus. On 5 June 2021, the General Administration of China Customs took emergency preventive measures due to the novel coronavirus nucleic acid positive test in the outer package samples of frozen small sardine imported from Pakistan. The customs of the whole country suspended the acceptance of the import declaration of this aquatic product manufacturer for one week from that day. Although these are only a tiny part of aquatic product epidemics in China, it seems that the outbreak of cold chain aquatic products (CCAP) has increased public consumption panic and consumer willingness to buy aquatic products is unclear. 

Many of the studies on purchase intention cited thus far have been from the perspective of the theory of planned behavior, in terms of attitudes, subjective norms, and perceived behavioral control to analyze the influence of consumer purchase intention [4,5]. However, there were few discussions of the purchase intention from the emotional response of consumers’ perspectives. Emotional responses have a more substantial impact on purchase intentions than instrumental elements such as attitudes [6]. Therefore, when faced with a significant health issue, emotional factors are more than worthy of consideration. However, there is relatively little empirical research on the influence of specific events (such as COVID-19) on the willingness to consume aquatic products. 

Therefore, this article will focus on the impact of specific events (COVID-19 outbreak involving CCAP) on the willingness to consume aquatic products from the perspective of micro-behavior. Based on the theory of planned behavior, this study represents a conceptual model of Chinese consumers’ desire to purchase CCAP. Then, the structural equation model was used to test the model and explore the mechanism of the factors influencing consumers’ purchasing intentions of CCAP under the epidemic situation by analyzing the data collected from the Ministry of Agriculture Marine Fish Industry Technical System. Studying such issues may practically contribute to the formation mechanism of consumers’ purchase intentions and public health in the epidemic context as well as identifying how to boost consumers’ confidence in the safety of aquatic products and ensure the healthy development of the aquatic product industry.

## 2. Literature Review and Hypotheses

The Theory of Planned Behavior (TPB) is a fundamental social psychology theory that explains personal behavior intentions from a micro perspective. TPB theory believes that attitudes, subjective norms, and perceived behavior control arise from various beliefs shaping an individual’s behavioral choices and leading to behavioral change [7]. Subjective norms reflect the influence of society on decision making. When making decisions, actors often consider the perception and expectations of significant people such as family, teachers, friends, and neighbors [8]. Perceived behavioral control is measurement standards that reflect how people can perform a specific behavioral choice. According to the theory of planned behavior, if people have a positive attitude towards consumption or think they can achieve their intentions, they will act with a cheerful willingness to consume [9]. Furthermore, if the individual has more favorable attitudes and subjective norms in behavior and more vital perceptual behavior control, the desire to carry out the behavior will be more assertive. As the theory is primarily concerned with what a person gains (or loses) from an action (weighing costs and benefits), it can be seen as a theory with a gain–goal framework approach [10]. The research of many scholars such as Leeuw et al. [11] and Li and Zhong [12] has proved that the theory of planned behavior has shown its effectiveness in different research fields, and the research on consumer purchasing behavior decisions has also been empirical. The attitude towards the safety of products is an essential factor affecting consumers’ purchases [13,14]. 

Moreover, the opinions of family and friends or various media reports on related events will affect consumers’ willingness to consume aquatic products [15]. Then, perceptual behavior control can manifest in consumers’ confidence in personal protection when entering vegetable markets, aquatic product wholesale markets, supermarkets, and other places, affecting consumers’ wishes [16]. As such, this paper proposes the following hypotheses based on the theoretical framework of planned behavior:

**Hypothesis** **1.**
*Consumers’ attitudes towards the safety of aquatic products (ATT) have a positive and significant impact on CCAP purchase intentions (BI).*


**Hypothesis** **2.**
*Subjective norms (SN) positively and significantly impact CCAP purchase intention (BI).*


**Hypothesis** **3.**
*Perceived behavior control (PBC) positively and significantly impacts CCAP purchase intention (BI).*


Consumers’ purchasing behavior is affected by many factors, which is the process of generating corresponding consumer behavior willingness and consumer behavior to meet their own needs. Unlike scholars in China who focus on quality and safety perceptions [13,17], foreign scholars pay more attention to consumers’ expected emotions [18,19]. The leading cause is that, although some scholars believe that attitude is one of the critical predictors in the theory of planned behavior, its predictive effect on behavior willingness has not achieved the expected outcome [20]. The possible reason for this situation is that attitudes emphasize more instrumental elements in the theory of planned behavior while ignoring emotional factors [21]. Although some scholars agree with this kind of view, through empirical research, some research results show that the influence of emotion on willing decision making is more significant than that of instrumental attitude [22]. However, other studies believe that there is no significant difference between the influence of emotion and instrumental attitude on willing decision making [23,24]. Therefore, based on previous research, this article considers the following hypotheses:

**Hypothesis** **4.**
*Emotional response to the health concern of CCAP (EM) has a positive and significant effect on attitude (ATT).*


**Hypothesis** **5.**
*Emotional response to the health concern of CCAP (EM) positively and significantly impacts subjective norms (SN).*


**Hypothesis** **6.**
*Emotional response (EM) has a positive and significant impact on perceived behavioral control (PBC).*


**Hypothesis** **7.**
*Emotional response to the health concern of CCAP (EM) positively and significantly impacts CCAP purchase behavior intention (BI).*


In previous studies, the mediating effects of attitudes, subjective norms, and perceived behavioral control were also considered. Shi et al. noted that attitudes have a mediating role in environmental influences on purchase intentions [25]. In a study of children in Hong Kong, China, the mediating role of subjective norms and perceived behavioral control was considered [26]. Matos and Krielow pointed out the mediating role of perceived behavioral control in environmental influences on e-goods purchase intentions [27]. Therefore, the following hypotheses are proposed:

**Hypothesis** **8a.**
*Attitude (ATT) partly mediates the relations between emotional response and CCAP purchase behavior intention (BI).*


**Hypothesis** **8b.**
*Subjective norm (SN) partly mediates the relations between emotional response and CCAP purchase behavior intention (BI).*


**Hypothesis** **8c.**
*Perceived behavior control (PBC) partly mediates the relations between emotional response and CCAP purchase behavior intention (BI).*


Extant literature shows that the moderating role of genders, ages, marital status, and family structures has been conclusive in the TPB framework [28]. For instance, 234 respondents selected from Jammu and Kashmir were asked to fill out the personal information questionnaire. The results showed that attitude on the intention toward online shopping was moderated by individual characteristics of age, gender, and marital status [29]. The different place of residence was also proved to be a moderator variable to affect the relationship between subject norms on purchasing behavior [30]. Moreover, research conducted in Vietnam found that purchase frequency could be a moderator variable [31]. Hence, the following hypotheses are proposed:

**Hypothesis** **9.**
*Gender moderates the relations between attitude and purchase intention (H9a); subjective norms and purchase intention (H9b); perceived behavior control and purchase intention (H9c); emotional response and attitude (H9d); emotional response and subjective norms (H9e); emotional response and perceived behavior control (H9f); and emotional response and purchase intention (H9g).*


**Hypothesis** **10.**
*Marital status moderates the relations between attitude and purchase intention (H10a); subjective norms and purchase intention (H10b); perceived behavior control and purchase intention (H10c); emotional response and attitude (H10d); emotional response and subjective norms (H10e); emotional response and perceived behavior control (H10f); and emotional response and purchase intention (H10g).*


**Hypothesis** **11.**
*Age moderates the relations between attitude and purchase intention (H11a); subjective norms and purchase intention (H11b); perceived behavior control and purchase intention (H11c); emotional response and attitude (H11d); emotional response and subjective norms (H11e); emotional response and perceived behavior control (H11f); and emotional response and purchase intention (H11g).*


**Hypothesis** **12.**
*Residence moderates the relations between attitude and purchase intention (H12a); subjective norms and purchase intention (H12b); perceived behavior control and purchase intention (H12c); emotional response and attitude (H12d); emotional response and subjective norms (H12e); emotional response and perceived behavior control (H12f); and emotional response and purchase intention (H12g).*


**Hypothesis** **13.**
*Purchase frequency moderates the relations between attitude and purchase intention (H13a); subjective norms and purchase intention (H13b); perceived behavior control and purchase intention (H13c); emotional response and attitude (H13d); emotional response and subjective norms (H13e); emotional response and perceived behavior control (H13f); and emotional response and purchase intention (H13g).*


The research framework of this article is shown in Figure 1.

## 3. Materials and Methods

This paper presents a questionnaire designed based on previous studies. After completing the questionnaire design, three experts (professors in economics, management, and semantics) were invited to review the questionnaire’s content. The unclear and easily misunderstood parts of the questionnaire were modified to increase their accuracy and quality and make it more understandable for respondents. The questionnaire was sent to the relevant practitioners in the marine product industry for consultation, and the problematic parts were revised. Finally, the questionnaire was pre-tested on a small scale in the National Marine Fish Industry System (NMFTS) of China’s Ministry of Agriculture. Thirty-five people from the 72 demonstration counties responded effectively, and some items were revised again to form the final questionnaire.

There are two parts of content included in the research questionnaire. The items in the first part (the demographic section) are about the personal characteristics of the interviewee, including age, gender, occupation, education, income, marriage, and residential area. The question items in the second part are about the respondent’s attitude toward the safety of aquatic products, subjective norms, perceived behavior control, emotional response to the health concern, and willingness to purchase aquatic products. The item design of the second part mainly refers to the scale design of Ajzen [32], Jain [6], and others (see Table 1). The Likert seven-level scale is adopted from 1 for “very disagree” to 7 for “very agree.”

The data were collected in China using a non-probabilistic sampling procedure. The surveyors are invited to complete the questionnaire in the designated online questionnaire system for the data collection process because using an online questionnaire can break geographical restrictions. The samples obtained are not limited to a single geographical location. The survey cost is reduced, and the recovery speed is faster, simultaneously. The samples were collected in June and July 2021 through channels such as the Integrated Experiment Stations (IES) of the NMFTS and the WeChat group of the research team. The system has established comprehensive stations and fixed observation points throughout China with provincial and county levels from top to bottom. We made full use of the observer resources in 72 cities and counties, and assigned them the task of finding enterprises and consumers in fixed observation sites to conduct research. We also conducted convenience sampling through our social channels. Some small gifts were given to participants who completed the questionnaire. The point-to-point organized assignment of tasks and bonuses for the respondents resulted in a high response rate. In the end, 841 sample data from 31 provinces, autonomous regions, and municipalities in China were obtained (excluding Hong Kong, Macao, and Taiwan, China), covering the six areas of Northeast China, North China, East China, Northwest China, Southwest China, and Central South China. After excluding the samples filled in randomly or with many missing items, a total of 783 valid questionnaires were obtained, with an effective response rate of the questionnaire being 93.10%. The demographic results of the respondents are shown in Table 2.

This paper applies the PLS-SEM (partial least squares structural equation model) research method. According to Anderson and Gerbing’s [34] method, it can be divided into two analysis stages. The first stage involves estimating the measurement model, including reliability and validity tests. The second stage is structural model analysis, mainly to assess and verify the path coefficient and explanatory power of the structural equation model to verify whether the constructed latent variables are reliable and effective, through these two stages and detect the causal relationship between latent variables to verify the hypothesis. The PLS-SEM method is called a second-generation statistical analysis technology [34], which discusses the path relationship between the potential variables. It can handle the model structure and measurement items simultaneously. In addition, PLS-SEM has relatively wide requirements for the normality and randomness of the data. Therefore, the relationship between the variables in the abnormal data distribution can be handled, and more robust results can be obtained.

Three matrix equations generally represent the structural equation model:(1)$$x=\Lambda_{x}\xi +\delta$$
(2)$$y=\Lambda_{y}\eta +\epsilon$$
(3)η=Bη+Γξ+ζ

In Formulas (1)–(3), x represents the exogenous observation variable vector, ξ is the exogenous latent variable vector, $\Lambda_{x}$ is the factor loading matrix of the exogenous observation variable on the exogenous latent variable, and δ is the residual vector of the exogenous observation variable. The endogenous observation variable vector is represented by y, η is the endogenous latent variable vector, $\Lambda_{y}$ is the factor load matrix of the endogenous observation variable on the endogenous latent variable, and ϵ is the residual term vector of the endogenous observation variable. Both B and Γ represent path coefficients, B represents the relationship between endogenous latent variables, Γ represents the influence of exogenous latent variables on endogenous latent variables, details the error term of the structural equation.

## 4. Results

### 4.1. Measurement Model Results

Measurement model testing is also external model testing, including reliability and validity testing. The reliability test index adopts Cronbach’s alpha coefficient and composite reliability (CR). Cronbach’s alpha coefficient is usually used to test the internal consistency of the measurement indicators. After the internal consistency test of the model, the Cronbach’s alpha coefficient of each latent variable is more significant than 0.6, and the combined reliability (CR) of each latent variable is more significant than 0.7 (see Table 3). Each latent variable’s observed factor loading is between 0.589 and 0.948. The above values comply with Fornell and Larcker’s [35] index requirements.

Then, it is necessary to test the validity of the model. The validity test includes the convergence validity test and the discriminant validity test. In terms of convergence validity and using factor loading and combination reliability to measure each item, it is also necessary to test each dimension’s average variance extraction (AVE) index. The structure has good convergence effectiveness if the AVE value is more significant than 0.5. As shown in Table 3, the average variance of the latent variables of the structure in this study is between 0.608 and 0.755, indicating acceptable convergence effectiveness.

The discriminant validity tests the difference between the measured variable and the different structural standards. The methods of Hair, Ringle, and Sarsted [36] were used to test the discriminant validity of the model samples and found that the AVE value of each latent variable was more significant than the square of the correlation coefficient between each latent variable. As shown in Table 4, it meets the index requirements proposed by Hair, Ringle, and Sarsted [36], indicating that the measurement model has acceptable discriminative validity.

### 4.2. Structural Equation Model Results

After completing the external model test, the internal model test is conducted, that is, the structural model test, to explore the path relationship between the latent variables. The model test results are shown in Table 5. Respondents’ attitudes towards the safety of aquatic products and subjective norms positively and significantly impact their willingness to purchase CCAP. H1 and H2 have been tested (ATT -> BI, β = 0.342, T statistic = 9.614; SN -> BI, β = 0.253, T statistic = 6.588). Emotional response to the health concern positively and significantly affects attitudes, subjective norms, and willingness to buy CCAP. Hypothesis 4, Hypothesis 5, and Hypothesis 7 are supported (EM -> ATT, β = 0.440, T statistic = 13.525; EM -> SN, β = 0.500, T statistic = 16.990; EM -> BI, β = 0.287, T statistic = 8.306). Perceived behavior control has no significant direct impact on CCAP purchase intention, and Hypothesis 3 is not supported (PBC -> BI, β = −0.014, T statistic = 0.489). Similarly, emotional response to the health concern has no significant effect on perceived behavior control, and Hypothesis 6 is rejected (EM -> PBC, β = −0.026, T statistic = 0.385).

Therefore, among the seven hypotheses proposed in this paper, except for Hypothesis 3 (*p* = 0.632) and Hypothesis 6 (*p* = 0.700), all other hypotheses satisfy the significance level test of *p* < 0.05. Consumers’ attitudes, emotional reactions, and subjective norms significantly impact the purchase intention of CCAP. Attitude has the most significant influence on purchase intention (0.342), followed by emotional effect (0.287), and subjective norms (0.253) have a weaker power. In addition, the emotional response has a significant positive impact on subjective norms and attitudes, which indicates that in addition to the direct effect of emotional response on purchase intention, there may be other intermediary effects on this.

### 4.3. Mediation Effect Results

To have a further in-depth understanding of the research model, this paper tests the mediation effect of the model. Preacher and Hayes [37] pointed out that the variance can express the accounted for (VAF), the proportion of indirect impacts to the overall effect when evaluating the mediation effect. There is no mediation effect when the VAF value is less than 20%. The VAF value is between 20% and 80%, indicating a partial mediation effect. When the VAF value is more than 80%, it means a full mediation effect. The results show that the influence of emotional reaction on consumers’ willingness to buy CCAP is realized through their attitudes toward aquatic product safety and subjective norms. The indirect effects are 0.151 and 0.127, respectively, and the VAF values are 34.47% and 30.68%, respectively. Therefore, it proves the partial mediation role of the attitude (H8a) and subjective norms (H8b). As for H8c, there is no mediation effect testing due to unsupported results of H3 and H6. Details are shown in Table 6.

### 4.4. Multi-Group Results

Moreover, consumer heterogeneity can be analyzed through a multi-group structural equation model [38]. Gender, marriage, age, residence, and monthly purchase frequency are considered moderator variables to investigate the influence of individual characteristics on consumers’ willingness to purchase CCAP under COVID-19. The results are shown in Table 7. It could be noted that there are no minors in the survey sample, and the age is divided into two groups: 40 years old and below (youth group) and 40 years old and above (middle-aged). The place of residence is divided into two groups: urban and rural. The monthly frequency of buying aquatic products is divided into eight times as a critical division, divided into eight times or less (low frequency), and eight times or more (high frequency). The verification results of the overall sample and the grouped sample are the same, as can be seen from the data in Table 5 and Table 7. However, there are differences in the analysis results of different individual characteristics in each grouping sample. Therefore, Hypotheses 9, 10, 11, 12, and 13 have been tested. Under COVID-19, personal characteristics will affect consumers’ willingness to buy CCAP.

## 5. Discussion

The current research identifies that the more optimistic the respondent’s attitude toward the CCAP epidemic, the stronger their ability to protect themselves and the higher their intention to purchase CCAP. In other words, the impact of the epidemic will be lower. The attitude towards aquatic product epidemics has the most significant direct impact on the respondents’ willingness to purchase CCAP, confirming that the respondents’ confidence in the safety of CCAP plays a vital role in decision-making consumer behavior [39]. Similarly, Latip et al. also found that attitudes toward food safety and health could influence personal intentions of purchasing [40].

The emotional response to the health concern also has explanatory power in affecting the willingness to consume CCAP. During the COVID-19 pandemic, individual emotions have constantly been fluctuating due to systemic factors, and whether to purchase CCAP is inevitably affected by mood. Subjective norms directly impact the willingness of CCAP consumption behavior, consistent with the research results of Shin et al. [41].

The direct effect of emotional response to the health concern on the respondents’ willingness to purchase CCAP is 0.287. The total impact of emotional response on the respondents’ desire to purchase CCAP through the path EM -> ATT -> BI is 0.438. Through the path EM -> SN -> BI, the total effect of emotional response on the respondents’ aquatic product purchase intention is 0.414. Therefore, it shows that the respondent’s emotional response to the health concern indirectly affects the willingness to consume through attitude and subjective norms. Furthermore, the more stable the respondent’s emotional response to the epidemic, the more positive the attitude, the more calmly they can look at other people’s views on the epidemic and the reports of various media, and the stronger their willingness to participate in consuming behavior. The notion parallels the suggestion of Son et al. that attitudes and subjective norms have a mediating effect in the process by which emotions influence purchase intentions [18]. Moreover, similar to our results, the finding of several studies suggested a positive relationship between emotional response, attitude, subjective norm, and intention [42,43].

This study shows no relationship between perceived behavior control and the respondents’ willingness to purchase CCAP and emotional response to the path of perceived behavior control, and none of them passed the significance test. However, this contradicts Chen and Huang and Shin et al., who concluded that perceived behavioral control could positively affect purchasing intention [41,44]. The possible reason is that under the normalization of the new coronavirus epidemic, even if the interviewees clearly understand the surrounding environment and their protection capabilities, their emotions are not the same as the anxiety and panic at the epidemic’s beginning [45]. Hence, the predictive and explanatory power of the dependent variable is insufficient.

Individual characteristics are heterogeneous in the willingness to purchase CCAP during the epidemic. Under the new coronavirus epidemic, married respondents are more likely to be affected by the opinions of family members, colleagues, and friends and various media reports. Moreover, similar to our results, the findings of several studies suggested a robust moderating effect of marital status [29,46] since the family is a complex system that acts on individual behavior and affects personal habits [47].

Next, from the residence’s perspective, there is a difference in the level of significance of emotional response and subjective norms. It indicates that respondents’ emotional response to the epidemic has affected their willingness to buy CCAP. Similarly, the opinions of family members, colleagues and friends, and various media reports are more likely to affect the purchase intentions of urban respondents. Chan and Chau came to a similar conclusion to ours, with different factors influencing the behavior of individuals in other regions [30]. The possible reason is that urban residents have access to more information sources and thus may receive more information.

Then, from the perspective of the monthly frequency of buying aquatic products, there are differences in the significance level of attitudes and emotional reactions. The more frequent the respondents who buy aquatic products, the more they will be affected by the opinions of family members, colleagues, and friends and various media reports. However, consumers with lower purchase frequency are more susceptible to attitudes, consistent with the research results of Won and Young [48].

In addition, there is a significant conclusion on the moderating effects on purchasing intention between subgroups of ages, the same as gender groups. In this study, both male and female, youth and middle consumers’ purchasing behavior can be affected, but this is inconsistent with previous results demonstrating that young males have a higher purchasing intention than older females [29]. This may be because the new coronavirus affects all age groups, men and women are similarly affected, and the epidemic in China is under control, so the willingness of CCAP consumption behavior does not differ significantly due to gender and age [45].

## 6. Conclusions

Drawn from the previous research, this paper proposed a new conceptual framework, including emotional responses, to analyze the factors affecting consumers’ aquatic product purchase behavior under COVID-19. The results show that attitudes toward the safety of aquatic products, emotional reactions to health concerns, and subjective norms have a significant positive impact on consumers’ willingness to purchase aquatic products. Especially, emotional reactions and attitudes are the main factors that affect consumers’ purchasing behavior in aquatic products. Another important conclusion of this paper is that emotional responses can influence consumers’ purchase intentions through attitudes and subjective norms. In addition, multi-group effects analysis has been performed to examine the performance in different demographic backgrounds. Specifically, middle-aged unmarried men are most likely to be influenced.

From the results, we provided essential information for the government to reconsider its policy to stabilize the consumer market. When dealing with food hygiene and health issues, the focus should remain on stabilizing consumer sentiment and disseminating knowledge through official media and other channels. In addition, based on our results for demographic moderating factors, the government should focus on middle-aged unmarried men as this group is vulnerable to food hygiene issues, and assistance is provided to them through the community. Furthermore, the new conceptual framework can be used as a reference for future studies. In this study, we have emphasized the factor of emotional response.

The first limitation of this study is the likelihood of common source bias due to self-reported data. In the future, we can use brain-imaging tools to decrease self-reporting bias in cross-sectional research. The second limitation is this study’s sample size (*n* = 783). Moreover, our results likely represent only the influencing factors of CCAP intention in China and cannot be generalized. Future research can use the conceptual framework of this study to investigate customer purchase behavior in different markets and other countries. It is suggested that qualitative research using open-ended questions may identify new factors influencing purchasing patterns and preferences over the pandemic to extend the literature.

## Figures and Tables

**Figure 1 ijerph-19-04903-f001:**
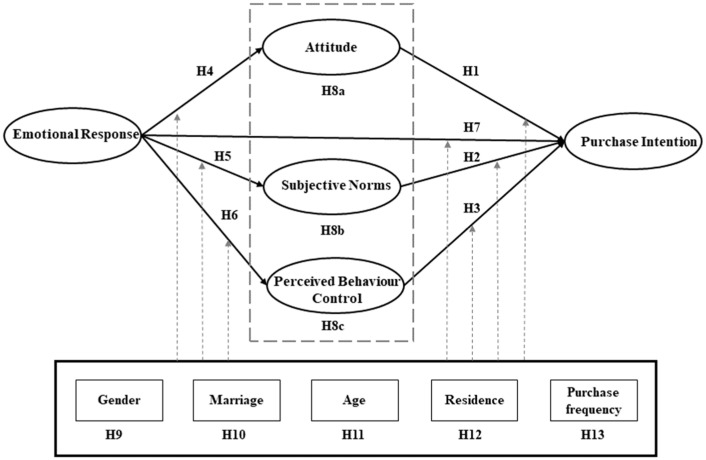
The theoretical model of consumers’ CCAP purchase intentions.

**Table 1 ijerph-19-04903-t001:** Measurement variables and sources.

Variable	Index	Items ^1^	Source
Attitude (ATT)	ATT1	Acknowledge that the epidemic is a factor in the safety of CCAP	(Ajzen and Kruglanskis [33])
	ATT2	Acknowledge that the epidemic has caused people to lack confidence in the safety of domestic CCAP	
	ATT3	Acknowledge that the epidemic has caused people to lack confidence in the safety of foreign CCAP	
Subjective norms (SN)	SN1	The opinions of my family will affect my purchase of CCAP	(Ajzen [32])
	SN2	The opinions of colleagues and friends will affect my purchase of CCAP	
	SN3	Reports from the official media will affect my purchase of CCAP	
	SN4	Traditional media reports will affect my purchase of CCAP	
	SN5	New media reports will affect my purchase of CCAP	
Perceived behavior control (PBC)	PBC1	Be sure to take personal protection when entering the aquatic product wholesale market	(Ajzen [32])
	PBC2	Be sure to take personal protection when entering a supermarket	
	PBC3	Be sure to take personal protection when entering the community	
	PBC4	Be sure to take personal protection when buying CCAP online and receiving goods	
	PBC5	Be sure to take personal protection when handling aquatic products	
Emotional response (EM)	EM1	Shocked by the CCAP epidemic	(Jain et al. [6])
	EM2	In the future, we will focus on the health impact of the CCAP outbreak	
	EM3	Concerns about the spread of the outbreak in the cold chain fish market and the health implications	
Purchase intention (BI)	BI1	The next month will reduce the consumption of CCAP	(Ajzen [32])
	BI2	The consumption of CCAP will be reduced in the next 3 months	
	BI3	The consumption of CCAP will be reduced in the next 6 months	
	BI4	If the consumption of CCAP is reduced, the consumption of aquatic products will be reduced	
	BI5	If the consumption of CCAP is reduced, the consumption of freshwater products will be reduced	

^1^ The items are deleted and revised from the original research version.

**Table 2 ijerph-19-04903-t002:** Basic statistical characteristics of respondents.

Variable	Index	Quantity	Proportion	Variable	Index	Quantity	Proportion
Gender	Male	369	47.13%	Regional distribution	North-east	112	14.30%
	Female	414	52.87%		North China	134	17.11%
Age	18~25	202	25.80%		East China	302	38.57%
	26~40	338	43.17%		Northwest	52	6.64%
	41~60	233	29.76%		Southwest	41	5.24%
	Over 60	10	1.27%		Central South	142	18.14%
Marriage	Married	510	65.13%	Average monthly income in the past year (yuan)	1000 and below	25	3.19%
	Unmarried	273	34.87%		1001~3000	68	8.69%
Have children	Yes	495	63.22%		3001~6000	174	22.22%
	No	288	36.78%		6001~9000	127	16.22%
Urban–rural distribution	City	641	81.86%		9001~15,000	160	20.43%
	Rural	142	18.14%		Over 15,001	229	29.25%

**Table 3 ijerph-19-04903-t003:** Confirmatory factor analysis of variables.

Latent Variable	Observed Variable	Factor Loading	Cronbach’s Alpha	CR	AVE
ATT	ATT1	0.851	0.763	0.864	0.679
	ATT2	0.858			
	ATT3	0.760			
SN	SN1	0.824	0.919	0.939	0.755
	SN2	0.842			
	SN3	0.883			
	SN4	0.914			
	SN5	0.879			
PBC	PBC1	0.864	0.923	0.938	0.751
	PBC2	0.847			
	PBC3	0.879			
	PBC4	0.830			
	PBC5	0.909			
EM	EM1	0.820	0.625	0.787	0.608
	EM2	0.586			
	EM3	0.811			
BI	BI1	0.926	0.912	0.936	0.749
	BI2	0.948			
	BI3	0.935			
	BI4	0.848			
	BI5	0.631			

Note: ATT = attitude; SN = subjective norms; PBC = perceived behavior control; EM = emotional response; and BI = purchase behavior intention.

**Table 4 ijerph-19-04903-t004:** Discriminant validity test.

	ATT	BI	EM	PBC	SN
ATT	0.679				
BI	0.364	0.749			
EM	0.194	0.319	0.608		
PBC	0.065	0.021	0.001	0.751	
SN	0.268	0.332	0.250	0.021	0.755

Note 1: The value on the diagonal is the average variance extraction, and the value below the diagonal is the square of the correlation coefficient between the latent variables. Note 2: ATT = attitude; SN = subjective norms; PBC = perceived behavior control; EM = emotional response; and BI = purchase behavior intention.

**Table 5 ijerph-19-04903-t005:** Model path coefficient and hypothesis testing results.

Path	Path Coefficient	T Statistics	*p*-Value	Hypothesis	Result
ATT -> BI	0.342 ***	9.614	0.000	H1	Supported
SN -> BI	0.253 ***	6.588	0.000	H2	Supported
PBC -> BI	−0.014	0.489	0.632	H3	Refused
EM -> ATT	0.440 ***	13.525	0.000	H4	Supported
EM -> SN	0.500 ***	16.990	0.000	H5	Supported
EM -> PBC	−0.026	0.385	0.700	H6	Refused
EM -> BI	0.287 ***	8.306	0.000	H7	Supported

Note 1: *** *p* < 0.001. Note 2: ATT = attitude; SN = subjective norms; PBC = perceived behavior control; EM = emotional response; and BI = purchase behavior intention.

**Table 6 ijerph-19-04903-t006:** Mediation effects testing results.

	Independent Variable	Mediating Variable	Dependent Variable	Direct Effect	Indirect Effect	Overall Effect	VAF	Result
H8a	EM	ATT	BI	0.287 *** (8.337)	0.151 *** (7.716)	0.438	34.47%	Supported
H8b	EM	SN	BI	0.287 *** (8.337)	0.127 *** (5.873)	0.414	30.68%	Supported

Note 1: *** *p* < 0.001. The value in () is the T value. Note 2: ATT = attitude; SN = subjective norms; EM = emotional response; and BI = purchase behavior intention.

**Table 7 ijerph-19-04903-t007:** Table of estimated parameters for grouping tests of different regulatory variables.

	Gender (H9)		Marriage (H10)		Age (H11)		Residence (H12)		Frequency (H13)	
	Male	Female	Yes	No	Youth	Middle	Urban	Rural	Low	High
	*n* = 350	*n* = 402	*n* = 495	*n* = 257	*n* = 516	*n* = 236	*n* = 618	*n* = 134	*n* = 579	*n* = 173
H1	0.28 ***	0.41 ***	0.35 ***	0.31 ***	0.41 ***	0.31 ***	0.35 ***	0.34 ***	0.35 ***	0.26 **
H2	0.29 ***	0.21 ***	0.27 ***	0.18 **	0.29 ***	0.23 ***	0.25 ***	0.27 **	0.22 ***	0.33 ***
H4	0.47 ***	0.36 ***	0.46 ***	0.39 ***	0.46 ***	0.44 ***	0.44 ***	0.45 ***	0.40 ***	0.53 ***
H5	0.52 ***	0.44 ***	0.50 ***	0.53 ***	0.49 ***	0.52 ***	0.45 ***	0.45 ***	0.51 ***	0.48 ***
H7	0.31 ***	0.26 ***	0.27 ***	0.34 ***	0.22 ***	0.33 ***	0.25 ***	0.25 **	0.31 ***	0.24 **

Note: ** *p* < 0.01; *** *p* < 0.001.

## Data Availability

The data presented in this study are available on request from the corresponding author.

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
