# Peer review of "Factors Influencing Consumers’ Purchase Intention on Cold Chain Aquatic Products under COVID-19: An Investigation in China"

_ijerph, 2022, doi:10.3390/ijerph19084903_

Round 1

Reviewer 1 Report

The paper includes the analysis of the research devoted to discover factors influencing customers purchase intensions on cold chain aquatic products. To solve this problem, a quantitative approach was used, i.e. exploratory factor analysis and then structural equations modeling. These are sophisticated and up-to-date statistical methods. They are used properly.

The logic of the paper, the methods, presentation and analysis are performed on a good level, in my opinion. Also, mediating effects and multi-group results were consider, which expands the scope of analysis widely.

My only preservation is that the "4. Discussion" is only discussion of the results obtained by the Authors. There are not any references to the similar studies carried out by other researchers, nor any comparison to these results. Please complete this lack of information, select a few similar researches and compare the results. This implies also from another disadvantege - the references used are somehow limited.

Two out of eight hypotheses were rejected, which is an interesting conclusion, much more interesting to the situation where all the hypotheses are supported. This shows that not all assumptions must be confirmed.

Reviewer 2 Report

In this article, the authors showed that attitudes (ATT) towards the safety of cold chain aquatic products (CCAP) and subjective norms (SN), have significant positive effects on the customer's purchasing behaviour intention (BI); the emotional response to the health concern (EM) of CCAP has a significant effect on SN, ATT, and BI. 
The emotional response to the health concern (EM) and subjective norms (SN) have significant positive impact on  Chinese consumer's willingness to purchase aquatic products. EM and ATT are the main factors that affect customer's purchasing BI in purchasing CCAP.

â‘  There are many abbreviations, such as SN, BI, EM ATT and so on. If possible, please show these abbreviations in the table, how to evaluate and what dose it mean for the readers of this article, in an easy-to-understand way.

â‘¡ Individual characteristics such as marriage, place of residence, and monthly purchase frequency significantly affect customer's purchasing BI of CCAP.
Please explain the reason why the individual characteristics affect customer's purchasing BI of CCAP more precisely.

Reviewer 3 Report

This review report aims to give some pointers to the paper’s authors entitled “factor influencing consumers’ purchase intention on cold chain aquatic products under Covid-19: an investigation in China”. Undoubtedly, the article shows several strengths as follows:

  • The research questions are relevant and adequate.
  • The questionnaire elaboration process seems rigorous.
  • The measuring instruments are fully described in their characteristics and bibliographical sources.
  • The use of statistical techniques is suitable and well defined.
  • The analysis of the results section is structured.
  • The contrast exercise of the hypotheses is explicit.
  • The conclusions encompass a summary and interesting limitations and future lines of research

Nevertheless, there are some shortcomings and, hence, there is room to improve the paper as follows:

  • There is not a particular section to reviewing the literature and the introduction section is too long. Therefore, let me suggest that the authors create two different sections: one for the introduction and another for the literature review.
  • The review of the literature and the introduction contents overlap one another. They are in the same section, but their contents are also mixed. So, please, pay careful thought to separate both kinds of content. Specifically, let me indicate as follows:
    1. Paragraphs 65-79 should be moved to the review of the literature section.
    2. Paragraph 80-94 encompasses research gap contents and theoretical contents. So, please, split these contents and while you leave the research gap contents for the introduction, but the theoretical contents in the review of the literature section. Similarly, lines 126-130 should be moved to the introduction, given that these are dealing with the research gap.
  • Although paragraphs 95-108 review the literature, it has not even a bibliographical reference. So, please, include bibliographical references in this paragraph.
  • In most parts, the hypotheses are well formulated and empirically contrastable. Nevertheless, they are not well supported due to a couple of reasons. First, there are not enough bibliographical references. For example, hypothesis 8 does not show a bibliographical reference. Second, the hypotheses are not explicitly supported. To address these problems, the authors should take two actions as follows:
    1. Please, support all the hypotheses by citing research works and scientists.
    2. Please, support the hypotheses separately so that you write up a particular paragraph with bibliographical references for each hypothesis.
    3. Hypothesis 8 formulation (lines 148-149) is confusing. What do you mean by “impact”? Is it a causal relationship or moderating effect? Please, use one of these terms. In addition, let me suggest that you break it down into five different hypotheses so that they turn out to be empirically and specifically contrastable.
  • H8 link is confusing and ambiguous in figure 1. If it indicates a moderating effect, the authors should draw an arrow from each moderating variable (gender, marriage, age, residence and frequency) to a mid-point of all the causal arrows in the model.
  • I guess the authors use a non-probabilistic sampling procedure by convenience, yet it is not said explicitly (lines 175-177). So please, use this technical lexicon.
  • It is unbelievable how the authors achieved such a high response rate (93.10%). I cannot see the ground for this success by reading lines 178-186. Typically, the response rate in online surveys is around 5-10%. Please, may the authors explain how they could reach this high percentage of responses?
  • It is reiterative to put the percentages in the text and display it in a table again (lines 187-196). I wonder if the authors might refer to this table in the text without repeating the same data in the text.
  • The subheading 3.3 (lines 278-291) looks interesting, but the authors do not relate this content to any hypothesis contrast. Please, let me suggest that the authors associate this section with the contrast of hypotheses.
  • Although the authors create a section dedicated to discussing (lines 309-356), there are only a few bibliographical references. So, please, make good use of this section to compare your obtained results to other papers’ authors to gain further insight into your evidence.
  • The conclusion section is devoid of managerial implications and lacks theoretical contributions. So please, come up with a practical impact for practitioners and highlight your paper's most original theoretical contribution.
  • Last but not least, give careful attention to some typos and misspelling problems. For example, “frequently” in line 35 and “concerns” in line 43. Please, proofread the paper.

I hope these comments help improve the paper and encourage the authors to move forward.

Author Response

Thank you so much for your suggestion, please see the attachment.

Reviewer 4 Report

The article on "Factors Influencing Consumers' Purchase Intention on Cold Chain Aquatic Products under Covid-19: An investigation in China" will be of interest to general readers of the journal. The investigation is relevant to the food system in other countries worldwide. However, some important considerations for the authors are highlighted in yellow in the uploaded manuscript.

L25-26: The sentence need to be revised to ensure clarity. 

L35: "frequently"

L41: Are you referring to a new variant of the coronavirus?

L93: marine

L96-100: Please revise the sentence and cite references

L200: "analyse"?

It is unclear from the discussion and limitation sections what and how the intermediary effects play a role in the purchase behaviour of consumers during the pandemic in China. In addition, it seems currently there is a rise in cases of coronavirus.

Round 2

Reviewer 3 Report

This is the second review and I have no choice but to acknowledge that the paper entitled “Factors Influencing Consumers' Purchase Intention on Cold Chain Aquatic Products under CovidCOVID-19: An investigation in China” has improved in several aspects. First, the introduction and review of the literature are separated. Second, there are new bibliographical references to support the hypotheses. Third, there are new hypotheses regarding specific sociodemographic characteristics. Fourth, figure 1 is clear and detailed. Fifth, the authors have improved their technical lexicon. Sixth, the sample is described in a table and the text is free from numerical and tiring to read information. Seventh, the hypotheses are better organised. Eighth, the discussion is enriched with bibliographical references and thus, it has gained insight. Finally, the conclusions section clarifies the theoretical contribution, comes up with managerial implications, acknowledges limitations and puts forward future lines of research.

Nevertheless, there is room to improve the paper as follows:

  • Some hypothesis formulations are dependent and do not refer to the model’s variables but other hypotheses (lines 177-181). Specifically, hypotheses 9a, 9b, 9c, 9d & 9e should refer to the variables rather than referring to other hypotheses. Please, revise these hypotheses’ wording statements.
  • It is hard to see how well-supported some hypotheses are because they do not have specific bibliographical references to support them. To be precise, hypotheses 8a, 8b and 8c need specific supporting paragraphs because they are pretty different from hypotheses 4, 5, 6 and 7. So, please, develop a specific review of the literature paragraph to support hypotheses 8a, 8b and 8c.
  • The questionnaire response rate (93%) looks fantastic by considering it was online. Although the authors explain to me it in their letter, they do not explain it in the paper. What is more, I do not see clearly why the respondents where so responsive. Please, identify the tips to reach such an amazing response rate and disclose it with brevity in the paper’s body text.
  • It is hard to see the final version of the paper given that it is not shown without the highlighted changes. So please, check the last version without highlighting the changes.

I hope these comments help improve the paper and encourage the authors to move forward.

Author Response

Dear reviewer,

Thank you for your careful review and good suggestion, which gives us a lot of inspiration from the last two rounds. For the response to round 2, please see the attachment.
